

# Dynamic precipitation phase partitioning improves modeled simulations of snow across the Northwest US

Bhupinderjeet Singh[1], Mingliang Liu[2], John Abatzoglou[3], Jennifer Adam[2], Kirti Rajagopalan[1]

[1]Department of Biological Systems Engineering, Washington State University, USA
[2]Department of Civil and Environmental Engineering, Washington State University, USA
[3]Management of Complex Systems, University of California, Merced, CA, USA

*Correspondence to*: Bhupinderjeet Singh (bhupinderjeet.singh@wsu.edu) or Kirti Rajagopalan (kirtir@wsu.edu)

**Abstract.** While the importance of dynamic precipitation phase partitioning to get accurate estimates of rain versus snow amounts has been established, hydrology models rely on simplistic static temperature-based partitioning. We evaluate model
skill changes for a suite of snow metrics between static and dynamic partitioning. We used the VIC-CropSyst coupled crop hydrology model across the Pacific Northwest US as a case study. We found that transition to the dynamic method resulted in a better match between modeled and observed (a) peak snow water equivalent (SWE) magnitude and timing (~50% mean error reduction), (b) daily SWE in winter months (reduction of relative bias from -30% to -4%), and (c) snow-start dates (mean reduction in bias from 7 days to 0 days) for a majority of the observational snow telemetry stations considered (depending on
the metric, 75% to 88% of stations showed improvements). However, there was a degradation in model-observation agreement for snow-off dates, likely because errors in modeled snowmelt dynamics—which cannot be resolved by changing the precipitation partitioning—become important at the end of the cold season. Additionally, the transition from static to dynamic partitioning resulted in an 8% mean increase in the snowmelt contribution to runoff. These results emphasize that the hydrological modeling community should transition to incorporating dynamic precipitation partitioning so we can better
understand model behavior, improve model accuracies, and better support management decision support for water resources.

## 1 Introduction

Snow processes are critical for accurate hydrological modeling, especially in regions where snowmelt significantly contributes to streamflow, such as the western United States (Li et al., 2017). A key aspect is an accurate partitioning of precipitation into rain and snow. This is because of the effects of rain-snow partitioning on land hydrology and climate—snow acts as temporary
storage for water, delaying its contribution to surface runoff, infiltration, and streamflow generation (Harpold et al., 2017). In contrast, rain contributes more rapidly to runoff and infiltration. Furthermore, variability in snow cover can influence the snow-albedo feedback mechanism, amplifying surface warming as snow cover decreases (Hall, 2004; Hall and Qu, 2006). Hence, the misrepresentation of precipitation phase in hydrology models can propagate through to inaccuracies in snow dynamics and subsequently affect snow and streamflow forecasts (Wang et al., 2019). Given that hydrology models are used to assess the
ability of the earth system to meet various demands such as irrigation, hydropower production, maintaining endangered fish



species, recreation, and navigation, the impacts of misrepresentation of the precipitation phase translate to these broad applications as well.

This is also important in a climate change context where a shift in the precipitation phase from snow to rain is projected to occur with a warming climate (Abatzoglou, 2011; Klos et al., 2014). The transition from snow to rain leads to
decreased snowpack (Kapnick and Hall, 2012; Mote et al., 2005; Pederson et al., 2011), impacts the streamflow magnitude and timings (Barnett et al., 2005; Fritze et al., 2011; Harpold and Brooks, 2018; Jepsen et al., 2016; Nayak et al., 2010), increases rain-on-snow flooding risk (McCabe et al., 2007; Musselman et al., 2018), and challenges our ability to make accurate water availability forecasts (Milly et al., 2008). Getting the precipitation partitioning right is essential to characterize these climate change implications as well.

Many hydrology models utilize simplistic static temperature thresholds to partition precipitation into rain and snow (Harpold et al., 2017). Recent studies (Jennings and Molotch, 2019; Marks et al., 2013; Wang et al., 2019) have demonstrated that this simplistic partitioning is inaccurate due to the dynamic variability in the temperature thresholds for partitioning (Jennings et al., 2018) as a function of other environmental factors. Therefore, using static uniform air temperature thresholds cannot accurately represent rain-snow partitioning across large spatial extents (Cho et al., 2022; Harder and Pomeroy, 2014)
and across time. A key environmental factor affecting the partitioning is relative humidity, with recent findings indicating that snowfall is more probable when precipitation is falling through an air mass with lower relative humidity due to evaporative cooling facilitated by latent heat flux (Froidurot et al., 2014; Gjertsen and Ødegaard, 2005; Jennings et al., 2018; Sun et al., 2019). Surface air pressure also influences precipitation partitioning, though to a lesser degree than air temperature and humidity (Jennings et al., 2018).

New partitioning methods have addressed these limitations. For example, Jennings et al. (2018) developed a bivariate rain-snow partitioning method as a function of relative humidity and air temperature. However, these methods are still not commonly integrated into hydrology models, and a large-scale evaluation of how this change in precipitation partitioning impacts modeled snow processes and agreement with observations is missing. Bridging this gap is important also because modeled snow metrics have high sensitivity to assumptions around precipitation phase partitioning (Sepúlveda et al., 2022;
Singh et al., n.d.; Sun et al., 2019). Our overarching goal is to integrate dynamic precipitation partitioning temperature thresholds into a hydrology model and evaluate its impact on the simulated snow processes and the contribution of snowmelt to streamflow. Focusing on the transition from a static to dynamic precipitation partitioning, the specific research questions addressed are (*a) Does this improve the match between modeled and observed snow magnitude and phenology, and by how much and where? and (b) How does the contribution of snowmelt to runoff change?*

To address our research questions, we utilize the VIC-CropSyst model (Malek et al., 2017), a coupled crop hydrology model where the hydrology aspects—including the snow model—come from a widely used land surface model, Variable Infiltration Capacity (VIC model version 4.1.2.e; Liang et al., 1994). The VIC hydrology model has been used worldwide in applications ranging from streamflow forecasting (Anghileri et al., 2016; Mazrooei et al., 2021; Ossandón et al., 2022; Singh et al., 2023), snow modeling (Bhend et al., 2012; Li et al., 2017; Schreider et al., 1997; Tang and Lettenmaier, 2010), climate



change impact assessment (Dang et al., 2020; Guo et al., 2009; Hengade et al., 2018; Krysanova and Hattermann, 2017; Treesa et al., 2017), and land-use change impact assessment (Carvalho et al., 2022; Garg et al., 2017; Hengade et al., 2018). However, despite its widespread use, VIC model implementations still rely on static temperature thresholds for precipitation partitioning. While we utilize the VIC-CropSyst model as a case study, the results are expected to be relevant more broadly in the context of other hydrological models, which, although vary in level of complexity and model structure, share a common set of similar

simplistic precipitation partitioning methods (Harpold et al., 2017 Table 2 comparing 24 hydrology models). Therefore, the insights from this study will be relevant in the broad hydrology modeling context.

        We utilize the Pacific Northwest US and Columbia River basin (CRB) as a case study region. This is an excellent model system for several reasons: (i) diverse climatic conditions ranging from arid, temperate, to cold climate classes (Beck et al., 2018), and (ii) exposure to a wide range of air temperature thresholds at which precipitation is equally split as rain and

snow, increasing from 0.6°C near the Pacific Coast to over 3.8° C in the Rocky Mountains (Jennings et al., 2018). Therefore, the knowledge gained from this region should be transferable to a broader set of regions.

## 2. Methodology

### 2.1 Study domain

We selected the Snow Telemetry (SNOTEL) stations in the Pacific Northwest region, encompassing the Columbia River Basin

(CRB) and coastal areas in the United States. The SNOTEL network was chosen as it provides daily snow water equivalent (SWE) data—defined as the depth of water that would result if the entire snow column melted—at sites spanning a broad geographic range, has a moderately long-term record of several decades, relatively consistent observational approaches across measurements, and has been widely used in other snow modeling efforts (Lute et al., 2022). The CRB is a vast snow-dominated watershed spanning multiple states in the US (Figure 1). The region heavily relies on snowmelt for water supply—50 to 80%

of the annual runoff is generated from the mountain snowpack (Li et al., 2017; Stewart et al., 2004), and has a diverse climate and topography. The area often experiences water scarcity, partly due to misalignment between water supply and demand timing (Hall et al., 2024), and there is high stakeholder interest in improved snow and streamflow forecasts.





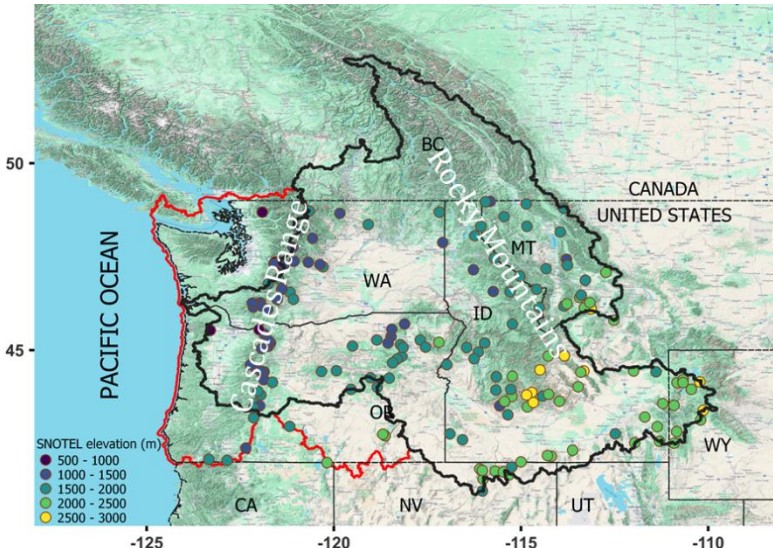

**Figure 1. Basin map of the Pacific Northwest, including the Columbia River basin (CRB) covering multiple states in the US and**
**Canada. The black boundary corresponds to the CRB. Points show the location and elevation of the SNOTEL stations used in this**
**study for snow process evaluation. The base map was created in QGIS v3.20.0 using the Google Terrain layer © Google Maps 2024.**

## 2.2 Input datasets

We utilized the calibrated VIC-CropSyst model and input data from Hall et al., (2024). The model requires gridded
meteorological inputs, soil and vegetation characteristics, topographic and land cover characteristics. The historical gridded

meteorological input is derived from the gridMET observational data product (Abatzoglou, 2013), which provides high-
resolution (~4-km, 1/24th degree) daily meteorological variables for the contiguous United States from 1979 to the present.
The gridMET data was spatially aggregated to 1/16th degree to align with the VIC-CropSyst model's resolution. Soil
information was obtained from the STATSGO2 (Staff, 2014) soil survey database, which comes from the United States
Department of Agriculture (USDA) Natural Resources Conservation Service (NRCS). For crop characteristics, two sources

were used: the Washington State Department of Agriculture's Land Use Layer for Washington State (WSDA, 2018) and the
USDA National Agricultural Statistics Service's Cropland Data Layer (USDA-NASS, 2016) for the rest of the contiguous
United States. The elevation and snow band information comes from Elsner et al. (2010) and Hamlet et al. (2010).

## 2.3 VIC-CropSyst model and calibration

The VIC-CropSyst model (Malek et al., 2017) is a coupled hydrology and agricultural model that integrates the VIC

hydrological model (Liang et al., 1994) with the cropping systems model (CropSyst; Stockle et al., 1994; Stöckle et al., 2003).
VIC-CropSyst runs at a 0.0625° spatial resolution and daily timestep considering water and energy balance to simulate the
infiltration, snow accumulation and melt, baseflow, surface runoff, and evapotranspiration. It does not simulate the sub-grid
lateral flow, as it is assumed to be negligible in large grid scales. There is a grid-level explicit representation of soil
characteristics, weather, vegetation, and crop types. It also statistically accounts for variations within each grid cell by



associating fractions of areas within the grid with multiple elevation bands, vegetation, and crop types. Using multiple elevation bands within a grid accounts for large variations in topography within a grid that can affect snow dynamics. Fluxes like surface runoff, baseflow, and evapotranspiration, are calculated for each vegetation class and aggregated for each grid cell based on the area covered by each class.

The snow sub-model calculates mass and energy balance to track snow accumulation and melting on the canopy and
ground. Andreadis et al. (2009) provide a detailed explanation of these processes. The snow melts when the net energy balance is positive, and if the excess water surpasses the liquid water holding capacity of the surface or deep pack layer, it is released to the soil as snowpack outflow. Conversely, if the energy balance is negative, any liquid water present freezes. The rain-snow partitioning algorithm relevant to this work is explained in section 2.4.

We utilized the VIC-CropSyst model calibrated as part of Hall et al. (2024). The model was calibrated for five soil
parameters and automatically calibrated based on the multi-objective complex evolution (MOCOM-UA) global optimization method. Calibration was performed in a nested approach that starts with calibrating grids for upstream stations and progressing to downstream stations until the entire watershed was calibrated. For this study, crop irrigation was turned off in simulations to capture natural settings. While the study primarily focuses on snow dynamics simulated by the VIC component of the model, the full VIC-CropSyst model was used because the streamflow calibration was performed on the coupled model version.
Overall, the evaluation of snow processes in the VIC-CropSyst model will be largely similar to that of the VIC model version integrated into the coupled model.

**2.4 Rain-snow partitioning algorithm**

In the VIC model, the precipitation partitioning into rain and snow is determined using a linear scaling method (Hamman et al., 2018), which we will refer to as the SRS (Static Rain-Snow) partitioning method throughout this study. This method sets
a lower-bound temperature (-0.5°C), below which all precipitation is classified as snow, and an upper-bound temperature (0.5°C), above which all precipitation is classified as rain. If the air temperature falls between the two bounds, the precipitation will be divided into rain and snow based on a linear interpolation (Cherkauer et al., 2003). These temperature thresholds are assumed to be constant across both space and time. Such simplistic assumptions are typical in most hydrology models as compiled for twenty-four well-used models in Harpold et al. (2017).

Within the VIC-CropSyst model, we implemented a bivariate rain-snow partitioning method (Jennings et al., 2018), which we will refer to as the DRS (Dynamic Rain-Snow) partitioning method in the rest of the paper. This method utilizes a binary logistic regression model to predict the probability of precipitation falling as snow or rain. The predictor variables in this model are 2-m air temperature (Ta, °C) and relative humidity (RH, %). The bivariate model is defined by the following logistic regression equation:


$$p_{snow} = \frac{1}{1 + e^{(-10.04 + 1.41Ta + 0.09RH)}} \tag{1}$$



where $p_{snow}$ is the probability of snow occurring. This model accounts for the non-linear relationship between air temperature, humidity, and precipitation phase, which is particularly critical near the freezing point where small changes in these variables can significantly affect whether precipitation falls as rain or snow.

## 2.5 Snow telemetry (SNOTEL)

Snow observations are from the Snow Telemetry (SNOTEL) network — a system of automated weather stations that provides
real-time data on snowpack and meteorological variables in the Western United States. The network was developed by the US Department of Agriculture's Natural Resources Conservation Service to support water management and planning activities in regions where snowmelt is a significant source of water supply. The stations are equipped with sensors that measure snow depth, SWE , air temperature, precipitation, and other variables related to hydrology and climate.

        We compare SNOTEL observations with VIC-CropSyst simulated results. In order to make the comparison as fair
as possible, we took a few steps. First, we only consider SNOTEL stations with an elevation difference between SNOTEL (point) and gridMET (grid) of less than ±150 meters to minimize elevation-related temperature biases. Second, the number of snowbands is set to one and its elevation is set to the SNOTEL station elevation. Finally, we excluded stations with more than 10% missing values per year for the water years 1997-2015. This resulted in 164 stations for our analysis.

## 2.6 Performance metrics

We evaluated the model's ability to capture multiple aspects of snow processes. We have eight performance metrics in total, comparing results between static and dynamic precipitation partitioning. The notation, brief description, and mathematical formulation associated with each metric are detailed in Table 1.

### 2.6.1 Peak SWE magnitude and timing

First, we assess snow magnitude using the SWE metric, defined as the depth of water that would result if the entire snow
column melted. This metric is crucial as it quantifies the amount of water stored in frozen form, which becomes available during spring melt. We assessed the peak SWE (maximum daily SWE in a water year) magnitude and timing.

        We calculated both absolute and relative biases to evaluate the performance of modeled peak SWE. Absolute bias measures the direct difference between simulated and observed values and can be expected to be higher in regions with substantial snow accumulation. We also calculated the relative bias to facilitate a meaningful comparison across regions with
varying peak SWE levels. We report the difference in days per water year to assess peak SWE timing.



### 2.6.2 Performance of the modeled daily SWE time series

To compare simulated daily SWE time series, we use two metrics: (i) Nash-Sutcliffe Efficiency (NSE) metric (Nash and Sutcliffe, 1970), a widely adopted standard for measuring the accuracy of hydrological models. (ii) percent relative bias (RB) for each location. For each location, NSE is calculated for the snow season (November through April in our study area) as follows:

$$NSE = 1 - \frac{\sum_{i=1}^{n}(SWE_{obs,i} - SWE_{sim,i})^2}{\sum_{i=1}^{n}(SWE_{obs,i} - \overline{SWE_{obs,t}})^2} \qquad (2)$$

where, n is the total number of observations, $SWE_{sim}$ and $SWE_{obs}$ are simulated and observed SWE respectively for a given day $t$, and $\overline{SWE_{obs,t}}$ is the mean of all observed daily SWE values. The value of NSE ranges from $-\infty$ to 1, with a value closer to 1 implying the best performance, and $\leq 0$ implying that the prediction is worse than using the long-term mean as the prediction.

We also calculated the relative bias (%) for daily SWE values on a monthly basis from November to April. The relative bias metric assesses the systematic deviation of the modeled SWE values from the observed SWE values. It is calculated as follows:

$$\%Relative\ Bias = \frac{\sum_{i=1}^{n}(SWE_{sim,i} - SWE_{obs,i})}{\sum_{i=1}^{n} SWE_{obs,i}} \times 100 \qquad (3)$$

Where $SWE_{sim,i}$ represents the simulated SWE value for the $i^{th}$ day and $SWE_{obs,i}$ represents the corresponding observed SWE value. A positive relative bias indicates an overestimation of SWE by the model as compared to observations, whereas a negative relative bias indicates an underestimation.

### 2.6.3 Snow phenology

We analyzed three snow timing variables, snow start (*SI*), snow-off (*SO*), and snow duration (*SD*). *SI* is the first day of the water year when the SWE is greater than or equal to 10 mm (Pan et al., 2003; Sobie and Murdock, 2022). This 10mm threshold avoids a false end corresponding to periods when SWE intermittently appears and melts due to temperature fluctuations at the start of the cold season. *SO* is the first day of 14 consecutive days when the SWE is zero. This condition avoids a false end. *SD* is calculated as the difference between *SO* and *SI*. These metrics are also calculated for simulations and observations.

### 2.6.4 Snowmelt contribution to streamflow

The contribution of snowmelt water to streamflow was quantified using a snowmelt tracking algorithm developed by Li et al. (2017). The algorithm monitors snowmelt water in surface water, soil, and the atmosphere. The fraction of streamflow originating from snowmelt ($f_{Q,snow}$) was determined using meteorological data, modeled surface, and subsurface runoff and baseflow fluxes, and water balance equations. The calculation of $f_{Q,snow}$ is as follows:

$$f_{Q,snow} = \frac{\sum Q_{snow}}{\sum Q} \qquad (4)$$



where $Q_{snow}$ represents the streamflow originating from snow, and $Q$ denotes the total streamflow. Streamflow at each time step $t$ is the sum of baseflow and surface runoff. Given that this metric is not compared with SNOTEL observations, the simulations are performed for a longer time period (1979 to 2015).

### 2.6.5 Relative model performance chart

For all the metrics discussed above, we provide a relative model performance (RMP) chart. When multiple models are compared, RMP can be quantified for each simulation instance and the results aggregated into an RMP chart. This is commonly used in comparing machine learning models (e.g., Gharsallaoui et al. (2024); Thapa et al. (2024)). For any simulation instance, the RMP of a specific model is the difference in that model's performance as compared to that of the best-performing model. If the model under consideration is the best-performing one, the RMP value will be zero. If not, the RMP provides an indication of how far the model's performance is from the best-performing model. RMP can be quantified for any performance metric and aggregated into an RMP chart where the X-axis corresponds to different RMP levels, and the Y-axis is the fraction of simulation instances with a particular RMP level. The closer a model's RMP curve is to the Y-axis and for longer, the better. The length of a model's curve exactly on the Y-axis (i.e., RMP equals zero) indicates how frequently the model is best performing, and the distance of the curve from the Y-axis indicates how much worse the model's performance is relative to the best model. In our case, the two models compared are the SRS and DRS implementations of precipitation partitioning, and the comparison instance is average performance values (e.g., mean absolute bias or NSE) for each SNOTEL station. So, the Y-axis values indicate what fraction of SNOTEL stations have specific RMP values for SRS and DRS partitioning.

**Table 1. Evaluation metrics used in this study**

| Notations | Short descriptions | Formulas |
|---|---|---|
| Peak SWE | The maximum amount of SWE on the ground per snow season (water year) | $max[daily\ SWE]$ *(mm)* |
| Day of peak SWE | Day of water year when peak SWE occurs | $Date\{max[daily\ SWE]\}$ |
| $f_{Q,snow}$ | Snowmelt driven streamflow | $f_{Q,snow} = \dfrac{\sum Q_{snow}}{\sum Q}$ |
| Snow-start | Start of snow season (water year) | $min(Date\{daily\ SWE > 10\ mm\})$ |
| Snow-off | End of snow season (water year) | After snow-start, $min(Date\{SWE = 0$ for 14 consecutive days$\})$ |
| Snow-duration | Length of snow season | Duration between snow-start and snow-off |





## 3. Results

### 3.1 Change in peak SWE magnitude by switching from SRS to DRS

Modeled average peak SWE across the 164 SNOTEL stations increased from 435 mm (SRS) to 534 mm (DRS), with an average increase of 27% (range 4% to 153%). The largest changes were observed over the stations with average daily temperatures within the 0-3°C range (Figure 2a). Within each temperature range, larger changes correspond to lower RH levels. Overall, the largest percent increase was observed when the average daily RH was between 60% - 70% and daily average temperature was between 1° and 3°C for wet days (precipitation > 1mm) between November and April. Lower RH

values than 60% would have larger changes, but we have limited data over these RH categories outside of very cold temperatures (Table S1). Spatially, the largest percentage changes were in the Cascades region (Figure 2b), where elevations are lower than the Rockies (Figure 1) and winter temperatures on days with precipitation are warmer, often falling within the 0-3°C range. Lower changes in peak SWE are observed in the colder temperature ranges (-5 to 0°C) across all RH levels (Figure 2a) because precipitation is already partitioned as snow at these temperatures, even with SRS partitioning.

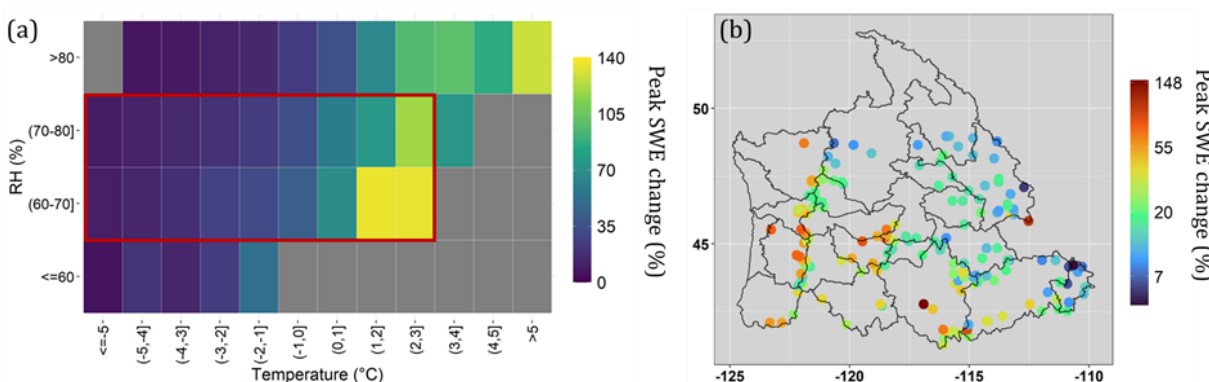


**Figure 2. Change in modeled peak SWE by switching from SRS to DRS partitioning method. (a) Change in peak SWE magnitude by average daily temperature and relative humidity categories for wet days (precipitation > 1mm) between November and April for all station-year combinations. The heat map only shows values in category groups with at least five station-years (see Table S1 in Supplementary Materials). Most data (~78%) are within the red box. (b) Spatial map showing each station's average change in peak**
**SWE (% change). The color bar for part b is on a log scale for visual clarity.**

### 3.2 Effect of switching from SRS to DRS on model-observation comparisons for all snow metrics

### 3.2.1 Annual peak SWE magnitude and timing

The SRS partitioning resulted in widespread underestimation (as compared to SNOTEL observations) of peak SWE (Figure 3a) with an average annual underestimation of 179 mm across stations. Switching to the DRS partitioning significantly reduced

this underestimation by around 50% (179mm to 87mm). While there is spatial variability in biases (Figures 3a and 3b) and changes in bias (Figure 3d), the DRS partitioning (Figure 3b) shows more regions with lower bias compared to the SRS method (Figure 3a). A similar pattern is observed in the relative difference between modeled and observed peak SWE (Figure S1a and




S1b in supplementary materials), with an average underestimation of 28% for the SRS partitioning, which reduces to 12% for the DRS partitioning.

In the Cascades range, there is a mix of over- and under-estimations of modeled SWE compared to observations with the DRS method. In contrast, the Rocky Mountain range shows more spatial consistency in the direction of the difference. The relative performance chart (Figure 3c) illustrates that the DRS method (black line) achieves better performance (for mean annual bias) than the SRS method (gray line) for 78% of the stations. The 22% of stations where the DRS partitioning worsened the agreement of modeled SWE with observations were those that had a high agreement to begin with, and the additional snow

with DRS partitioning degraded the simulation agreement with observations, often resulting in an overestimation of peak SWE (darker shades of blue in Figure 3b).

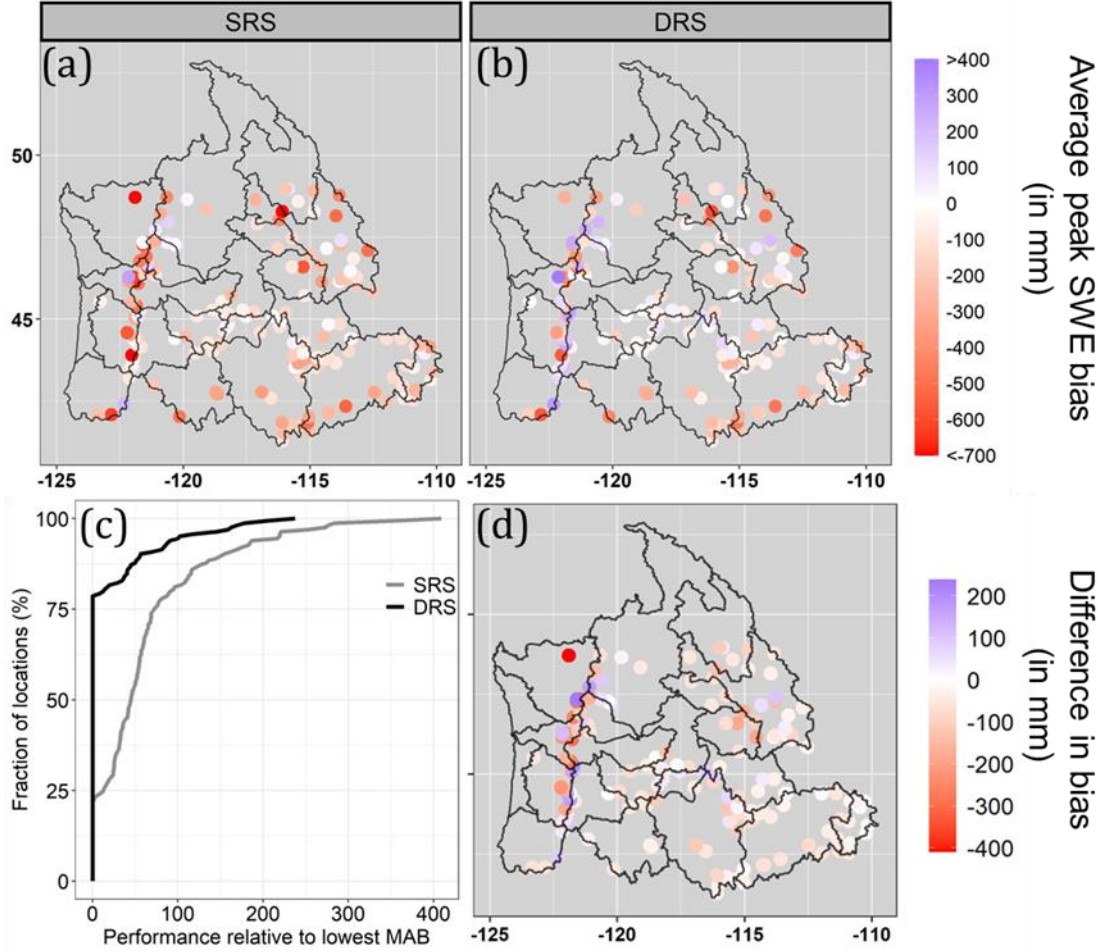

**Figure 3. The average bias in peak SWE (modeled - observed) over SNOTEL stations and the changes in bias between SRS and DRS precipitation partitioning methods. (a) and (b) Bias in peak SWE magnitude for SRS and DRS partitioning, respectively. (c) Relative**
**model performance (RMP) chart: The Y-axis is the fraction of stations for which a particular RMP is achieved, and X-axis is the difference between each model's mean absolute bias (MAB) and the best-performing model's MAB. The closer a model's curve is to the Y-axis and for longer, the better. The length of a model's curve exactly on the Y-axis indicates how frequently the model is best performing, and the distance of the curve from the Y-axis indicates how much worse a model's performance is relative to the best**





**model. See methods section 2.6.5 for more details on interpretation. (d) Change in absolute bias (|DRS| − |SRS|). The negative values (red) indicate where DRS partitioning reduced the bias, and positive values (blue) indicate where the bias got worse with the implementation of DRS partitioning.**

DRS improves model performance on the timing of peak SWE compared with SRS (Figures 4a and 4b). When using SRS partitioning, the mean timings of simulated annual peak SWE are 16 days earlier than observations on average. However, implementing DRS partitioning reduces this difference to 8 days, improving the alignment of the timings of model-simulated peak SWE with observed data by about 50%. Approximately 86% of the stations showed improvement in the timings of peak SWE with the DRS method (Figure 4c). Even where the DRS method is not the best performing, its performance remains relatively close to that of the SRS method, with a median difference of 2 days and a maximum difference of 10 days. In contrast, when the SRS method is not best performing, it can be worse than the DRS by a median of 9 days and as large as 50 days for some locations (Figure 4c), highlighting its potential limitations.

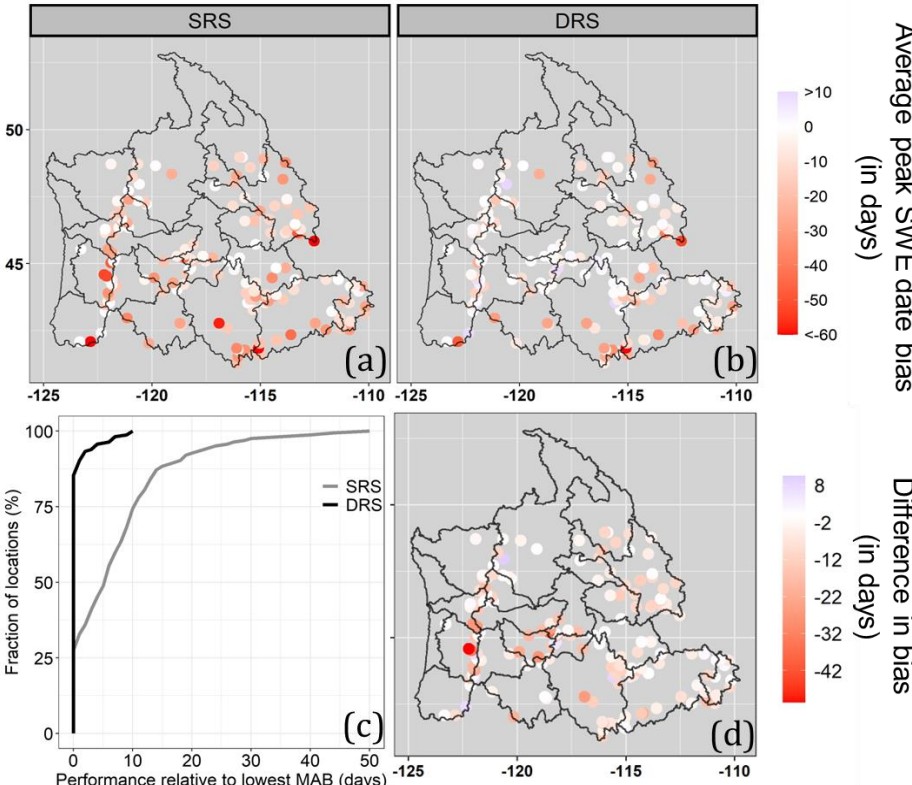

**Figure 4. The average bias in peak SWE timing (modeled - observed) over SNOTEL stations and the changes in bias between SRS and DRS partitioning. (a) and (b) Bias in peak SWE timing for SRS and DRS partitioning, respectively. (c) Relative model performance (RMP) chart. See Figure 3 caption and methods section 2.6.5 for more details on interpretation. (d) Change in absolute bias (|DRS| − |SRS|). The negative values (red) indicate where DRS partitioning reduced the bias, and positive values (blue) indicate where the bias got worse with the implementation of DRS partitioning.**





### 3.2.2 Daily SWE magnitude

DRS partitioning performs better for daily SWE, with an average NSE of 0.63, compared to 0.55 for SRS partitioning (Figures 5a and 5b). Additionally, the DRS method achieved high accuracy (NSE ≥ 0.66) at half of the stations, compared to only one-third for the SRS method. Overall, the DRS method improved results at 76% of stations, with an average NSE increase of 0.3 (Figure 5c). The number of stations with negative NSE values reduced from 33 (SRS) to 20 (DRS) (Figures 5a and 5b gray points).

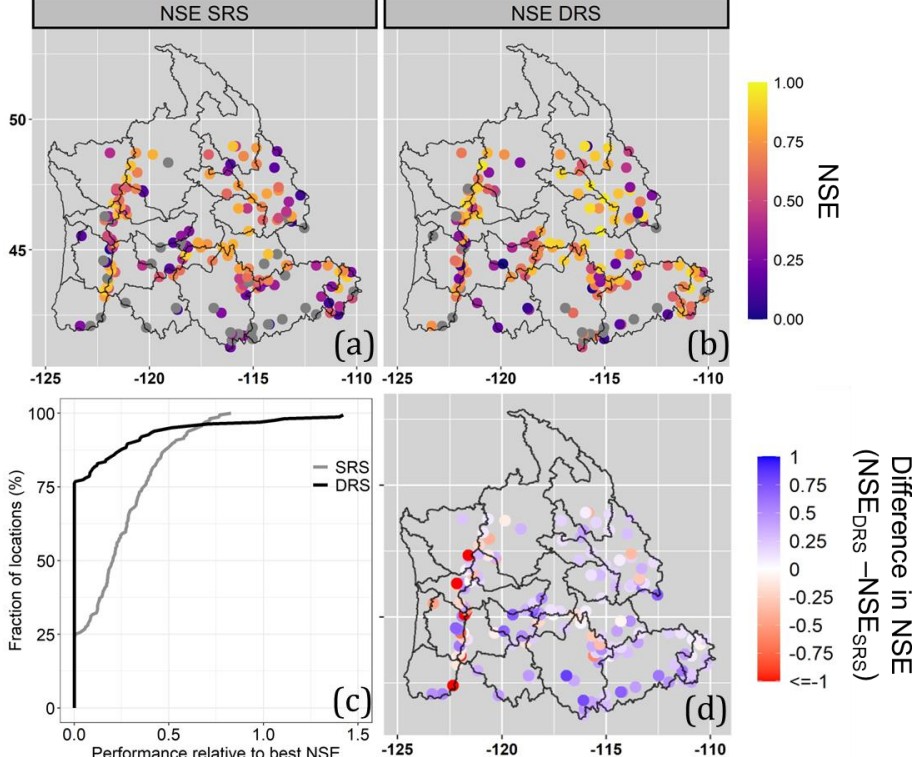

**Figure 5. Model performance (NSE) of daily SWE for SRS and DRS partitioning for the snow season (November through April) and the difference in performance. (a) SRS method (b) DRS method. The gray color points in a) and b) indicate the stations with NSE smaller than 0. (c) Relative model performance (RMP) chart. See Figure 3 caption and methods section 2.6.5 for more details on interpretation. (d) Change in NSE (DRS – SRS). The negative value (red) color indicates where DRS method degraded the NSE, and positive value (blue) indicates where it improved the NSE.**

In terms of mean relative bias, similar to the NSE metric, there were substantial reductions (-30% to -4% bias) across the snow season (i.e., November-April) when transitioning from SRS to DRS partitioning. The bias with the DRS method is nearly zero during mid-winter (December to Feb). However, bias persists in the warmer shoulder months of November (start of snow season) and April (end of snow season). In November, the relative bias reduced from ~45% (SRS) to -9% (DRS) (Figure 6). In April, the relative bias reduced from -46% (SRS) to -23% (DRS). Overall, the DRS method reduced the relative bias in approximately 75% of the stations, with an average reduction of 19% across these stations.




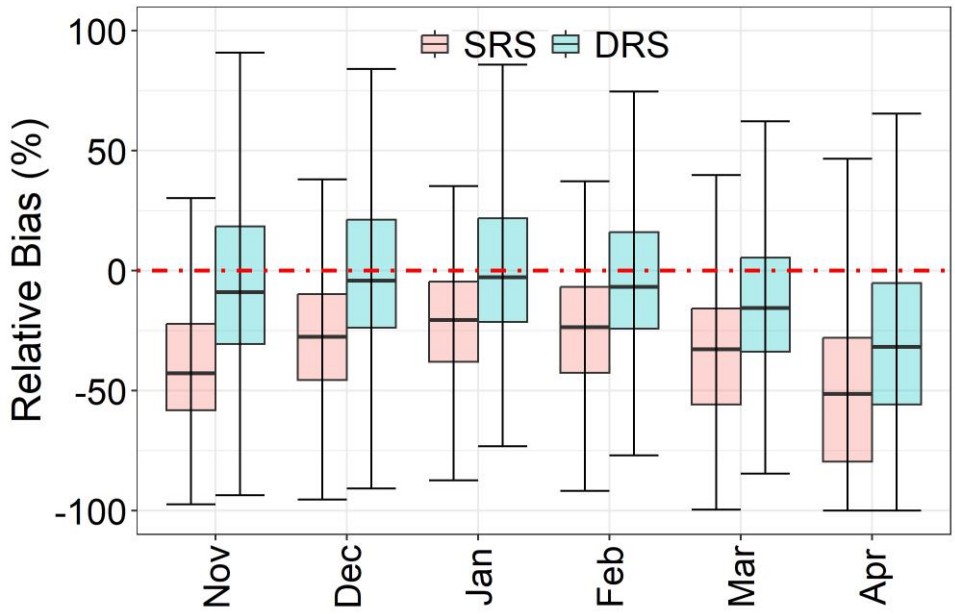


**Figure 6. Box plot of average percent relative bias of mean daily SWE over each winter-season month (November-April) for the SRS and DRS partitioning method. The lower and higher edges of the box represent the first (Q1) and third (Q3) quartiles, respectively, indicating the interquartile range (IQR). The black line inside the box represents the median value. The whiskers extend to 1.5 \*IQR from the lower and upper quartiles, while outliers beyond this range are not shown.**

### 3.2.3 Snow phenology

In examining the snow phenology metrics, we observed distinct differences in performance between DRS and SRS partitioning across metrics. In terms of snow-start, the mean bias is almost fully reduced from ~7 days (SRS) to ~0 days (DRS) (Figures 7a and 7b). The RMP chart also shows that the DRS method outperforms SRS across 73% of stations (Figure 7c). However, the comparison switches for the snow-off metric with the mean error worsening from ~0 (SRS) to ~11 days (DRS) (Figures 7d and 7e). Note, however, that with SRS partitioning, very few stations have 0 average error, but positive and negative differences across stations cancel each other out to give an average mean error of ~0 (Figure 7d). Since overall snow duration depends on snow-off and snow-start, the increased snow-off errors translate to higher snow-duration errors for the DRS method. The DRS method shows an average overestimation of 13 days (Figure 7h), whereas the SRS method indicates an underestimation of approximately 4 days (Figure 7g).






**Figure 7. The average bias and change in bias for snow phenology metrics (modeled - observed) for SRS and DRS partitioning. The columns represent the three-phenology metrics: snow-start (a, b, c), snow-off (d, e, f), and snow-duration (g, h, i). The spatial maps in the first row (a, d, g) and second row (b, e, h) show average bias in snow phenology metrics using SRS and DRS partitioning methods, respectively. The third row (c, f, i) shows the Relative Model Performance (RMP) Chart. See Figure 3 caption and methods**

**section 2.6.5 for more details on interpretation.**

### 3.3 Contribution of snowmelt to streamflow

We used the snowmelt tracking algorithm (Li et al., 2017) to analyze the impact of switching from SRS to DRS partitioning on the estimations of snowmelt contribution to streamflow (i.e. $f_{Q,snow}$). The average snowmelt contribution increased from 56% (SRS) to 64% (DRS) (Figures 8a and 8b). The difference map (Figure 8c) highlights that this direction of increase is

largely consistent across all stations (blue values). The average increase in $f_{Q,snow}$ values is approximately 8%, with some locations experiencing up to an 18% increase. The highest increase is observed in lower elevation, warmer areas, particularly



in the Cascades and the southern regions of the Rockies. Given that observations of $f_{Q,snow}$ do not exist, a model-observation agreement comparison is not possible.

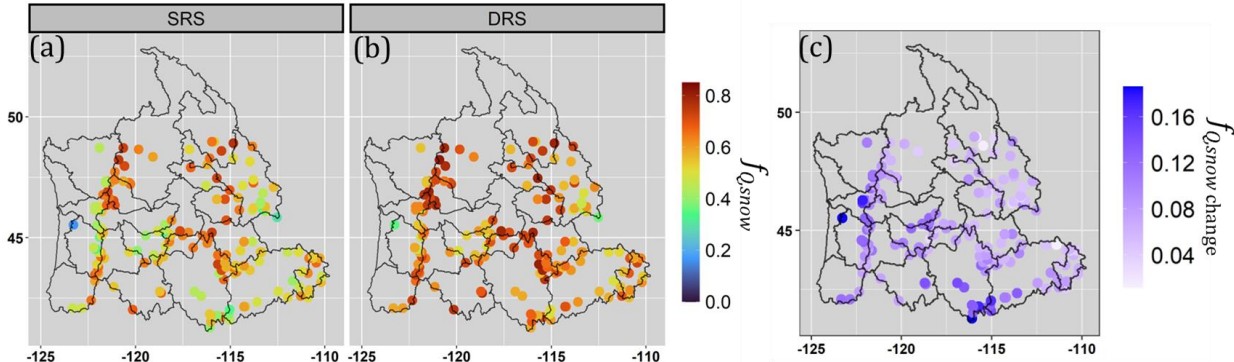

**Figure 8. Spatial map of average $f_{Q,snow}$ for the period 1979-2015 (a) SRS (b) DRS (c) change in $f_{Q,snow}$ by switching to DRS from SRS method (i.e., $f_{Q,snow,DRS} - f_{Q,snow,SRS}$).**

**3.4 Drivers of changes in peak SWE error (model – observations) between SRS and DRS**

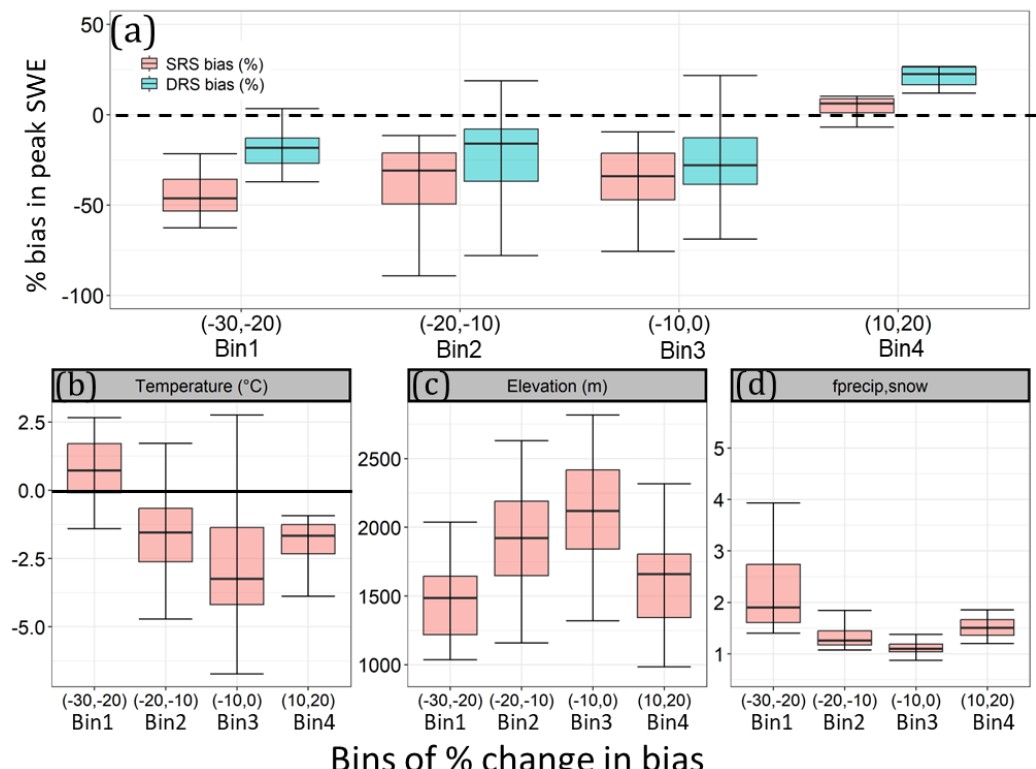

**Figure 9. Percent bias in peak SWE and average temperature, elevation, and fraction of accumulative precipitation to peak SWE for bins of % change in bias in increments of 10% change in bias. Only bins with at least 8 stations are displayed and this corresponds**






**to 84% of the stations. Bins 1, 2, and 3 with negative changes in bias correspond to performance improvements with DRS. Bin 4, with a positive change in bias corresponds to degradation in performance with DRS. For each bin, the distribution of four aspects are provided across the stations within each bin. These include (a) biases of SRS and DRS methods, (b) average daily temperatures wet days (precipitation > 1mm) during the snow season (November-April), (c) elevation, and (d) the ratio of cumulative precipitation from October 1st until peak SWE has been attained and the peak SWE value from SNOTEL observations. This is to get a sense of potential precipitation undercatch issues.**

The 164 SONTEL stations were binned into groups based on their percent change in average peak SWE bias (in increments of 10%) when switching from SRS to DRS (|DRS %bias| - |SRS %bias|). Only bins with at least 8 stations were analyzed resulting in four bins (Figure 9). Bins 1, 2, and 3 comprise stations with improvements in peak SWE performance with the

DRS methods with decreasing levels of improvements (Figure 9a). They also have a decreasing pattern of average temperatures (Figure 9b), increasing pattern of elevations (Figure 9c), and decreasing patterns of the precipitation to SWE ratio (Figure 9d). Bin 4 comprises stations with degradation in performance. These stations had low errors to begin with even with the SRS method (Figure 9a), and additional snow with the DRS method resulted in an overestimation of peak SWE. While this set of stations had similar elevation ranges as Bin 1, where the performance improvements with DRS were highest, stations in Bin 4

are relatively colder than Bin 1.

Bin 1 comprises stations with relatively large biases with the SRS method and with the largest improvements with the switch to the DRS method. There are generally low-elevation stations (Figure 9c) with daily average snow-season temperatures in the 0-3 °C range and relatively larger precipitation to SWE ratios. Bin 3 comprises stations that had biases with the SRS method that could not be improved much with the DRS method. These are relatively colder (Figure 9b), high

elevation (Figure 9c) areas where the SRS method itself should have handled the precipitation partitioning well. However, the ratio of precipitation to peak SWE is very low, indicating that this set of stations might have potential issues with precipitation observations and there is just not enough precipitation to translate to observed SWE values by any model.

## 4. Discussion

The dynamic partitioning resulted in more snow than the static method across all 164 SNOTEL stations analyzed, similar to

previous findings across the western US (Jennings and Molotch, 2019). The estimation increases in snow magnitude were more prominent when temperatures ranged between 0-3°C and in lower RH conditions. This temperature range is consistent with other work that has demonstrated that regions most impacted by changes in rain-snow partitioning are typically those with winter temperatures near freezing 0°C (Ding et al., 2014; Jennings et al., 2018; Nolin and Daly, 2006). More snow in low RH conditions also makes physical sense as falling snowflakes experience evaporative cooling in dry air, allowing it to remain

frozen at higher temperatures (Harpold et al., 2017).

Moreover, this additional snow (from dynamic partitioning) better aligned modeled SWE with observations in most cases. Jennings and Molotch, (2019) demonstrated that the DRS method resulted in better agreement of modeled daily SWE and snow depth for eleven research stations. This work demonstrates the same for a more comprehensive set of 164 SNOTEL stations across the western US and for a larger set of metrics: magnitude and timing of peak SWE, overall daily SWE, and





snow phenology). There are three exceptions that resulted in either a degradation of model performance or in a limited change
in performance by switching from SRS to DRS partitioning. The first relates to degradation in estimates of snow-off dates,
which also resulted in a degradation of snow duration estimates. This is likely a result of the larger relative bias in simulated
SWE in the shoulder month of April (Figure 6). A possible explanation is that snowmelt plays a significant role in determining
late-season SWE, and imperfections in the model melt dynamics — an aspect that would not be resolved by transitioning from

a static to dynamic precipitation phase partitioning. The second exception is that even for the metrics that by and large see
model performance improvements, degradations are observed in some stations (12% to 25% of stations, depending on the
metric) (Figures 3c, 4c, 5c, and 7c). These are likely stations where the SRS partitioning itself was working well and additional
snow with a switch to a DRS method resulted in overestimation of SWE (Figure 9a Bin4). The third exception is a set of
stations with minimal changes in simulated metrics between SRS and DRS partitioning. These stations typically correspond

to higher elevations—with colder winter temperatures (Figure 9b)—where winter precipitation is partitioned as snow in both
methods. The likely reason for persistent larger errors in some of these stations are precipitation undercatch issues (Figure 9d)
which are unrelated to precipitation phase partitioning.

Discrepancies between modeled and observed SWE have been reported by multiple studies (Broxton et al., 2016;
Cho et al., 2022; Hamlet et al., 2005; Islam and Déry, 2017; Pan et al., 2003; Wang et al., 2019). However, normally they have

been attributed to errors in meteorological inputs due to issues such as precipitation undercatch (e.g., Cho et al., 2022; Hamlet
et al., 2005; Islam and Déry, 2017; Pan et al., 2003). This work, however, demonstrates that model improvements in-and-of-
itself can reduce a large fraction of errors (e.g ~50% average improvements in peak SWE across 82% of stations with
improvements). Moreover, addressing this issue will help us better attribute the remaining error to other sources, including
issues with meteorological data or other processes like snowmelt dynamics.

The importance of accurate rain-snow partitioning in hydrological models becomes even more critical under climate
change simulations, as the partitioning directly influences runoff generation, soil moisture, groundwater recharge, and snow
albedo feedback (e.g., Harder and Pomeroy, 2014; Jennings and Molotch, 2019). SRS partitioning can lead to misleading
interpretations of higher levels of reduction in snowpack with warming (Harpold et al., 2017). This is even more relevant in
regions where decreasing trends in RH have been projected (Byrne and O'gorman, 2016; Harpold and Brooks, 2018) given

that the differences in SWE estimated by SRS and DRS methods are larger under lower RH levels (Figure 2a). Additionally,
a higher contribution of snowmelt to runoff with DRS partitioning (8% average) (Figure 8c) could result in fewer projected
transitions from snow-dominant to transitional or rain-dominant watersheds, though this would need further investigation.
This transition is a key focus area of several studies (e.g., (Foster et al., 2016; Li et al., 2017; Schnorbus et al., 2014; Tohver
et al., 2014)) and transitioning to a DRS partitioning can provide a more accurate picture of these impacts.

Some limitations in our approach can affect interpretations. A key aspect relevant to any work comparing modeled
snow to point observations is a scale mismatch. Coarser resolution of gridded input datasets to models fail to capture the fine-
scale variability seen in point measurements and may smooth out extremes and variability that are crucial for accurate snow
modeling (Lundquist et al., 2015). Additionally, modeled processes are prioritized to be scale-appropriate (Archfield et al.,



2015) and the key processes might be different at the observational point scale of SNOTEL stations. While we minimize the
impact of scale mismatch by restricting our analysis only to grids where the elevations are close to the corresponding SNOTEL station, and running the model using the SNOTEL elevation, the comparison cannot be perfect.

Moreover, the SNOTEL observations themselves have inaccuracies for multiple reasons. This includes precipitation undercatch, particularly during snowfall events (e.g., Pan et al., 2003; Scalzitti et al., 2016) resulting from wind effects and limitations of precipitation gauges, leading to an underestimation of precipitation and subsequently modeled SWE. Another
source of error is snowdrift—which can cause snow to accumulate unevenly across landscape features (Meyer et al., 2012; Sun et al., 2019). Lack of appropriate maintenance at stations (e.g., keeping clear of vegetation) and extraneous influences like animal activity can also contribute to errors (Meromy et al., 2013). These issues with observations create challenges in attributing differences between modeled data and observations as model error.

Finally, our dynamic partitioning approach only integrates the effect of the key driver, relative humidity. More
comprehensive partitioning approaches that integrate other physical factors or region-specific modeling of relationships using physical and artificial intelligence approaches can help improve the partitioning further.

## 5. Conclusions

Our findings demonstrate that employing more dynamic representations of rain-snow partitioning temperature thresholds can lead to significant bias reductions in modeled snow magnitude and timing (~50% reduction on average), and align modeled
output more closely with observations. Most hydrology model applications continue to rely on simple static precipitation phase partitioning. Our results underscore the need for the hydrology modeling community to adopt dynamic methods of precipitation partitioning (e.g., Jennings et al., (2018)) as a routine practice. As a key sensitive parameter of hydrology models with respect to snow processes (Sepúlveda et al., 2022; Singh et al., n.d.), this transition would be critical for a better understanding of model behavior, improvements in model accuracies, and ultimately better support of water resources management. This is
especially important also for realistic climate change assessments that quantify reductions in snowpack or transitions of watersheds from snow-dominant to rain-dominant regimes in a warming climate.

*Code availability.* The VIC-CropSyst model code is available in the following GitHub repository (https://github.com/mingliangwsu/VIC-CropSyst-Package.git). The codes used in this study are available from the
corresponding author upon reasonable request.

*Data availability.* Input data are all from public sources that are referenced in the methodology section. Any intermediate model output will be made available by the authors upon request.



*Author contributions.* **Bhupinderjeet Singh**: Writing – review and editing, Writing – original draft, Methodology, configured the VIC model, conducted simulations, analyzed results, and created the figures. **Mingliang Liu**: Writing – review and editing, Methodology, insights into results. **John Abatzoglou**: Writing – review & editing, insights into results, **Jennifer Adam**: Writing – review & editing, insights into results, **Kirti Rajagopalan**: Writing – review & editing, Writing – original draft, Supervision, Resources, Project administration, Methodology, Conceptualization.


*Competing interests.* The contact author has declared that neither they nor their co-authors have any competing interests.

*Financial support.* This work was supported by the USDA-NIFA/NSF AI Research Institutes program, under award No. 2021–67021–35344.

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
