# Peer review of "Dynamic precipitation phase partitioning improves modeled simulations of snow across the Northwest US"

_EGUsphere, 2024_

## Author Comment (AC1)

**Reviewer #1**

Thank you for the feedback and suggestions for improvements. We agree with all the suggestions, including the suggestion for additional analysis related to basin-level peak SWE/Q comparisons. All suggestions can be incorporated in the revised manuscript. Specific responses are provided in detail below in red font.

**Summary**

This manuscript examines the implications on simulated snowpack conditions associated with changing a precipitation partitioning model (partitioning precipitation into rain and snow fractions), from an approach considering only contemporary air temperature to an approach using both contemporary air temperature and humidity. The authors carry out a suite of numerical experiments in the Pacific Northwest region of the United States using the two alternative partitioning schemes and compare the approaches to each other and to observational data at USDA SNOTEL observation sites. Simulations and observations are compared on the basis of peak snow water equivalent (SWE), the timing of peak SWE, snow phenology (that is, the onset, conclusion, and duration of seasonal snow cover), and the fraction of runoff from snow at SNOTEL sites within the Columbia River Basin. Overall, the switch to a bivariate precipitation partitioning approach yields improvements in simulated snow conditions with respect to observations. Where there is degradation, the authors effectively argue that the degradations are likely the influence of factors beyond those that would be influenced by the change in partitioning approach. The paper is of interest to the readership of HESS and makes important, albeit primarily methodological, contributions to snowpack and hydrologic modeling in snow-dominated regions. I believe that it can be published with only minor revisions.

**Major Comments:**

a) I can surmise why VIC-CropSyst is being used in this study, instead of VIC without the CropSyst model coupled. However, some readers might be left wondering why VIC-CropSyst is being used when, for example, value additive aspects of the crop model (e.g., dynamic crop yields, etc.) are not being examined and when the comparisons with SNOTEL observations are primarily in locations where there is little or no cultivation. As such, I would strongly encourage the authors to provide some additional context for using VIC-CropSyst for readers. For example, if the work summarized in the manuscript is part of larger and/or ongoing efforts to develop more sophisticated regional projections of how climate change might affect agriculture in the region, it would be helpful for readers to know.

The use of VIC-CropSyst in this study, rather than the standalone VIC model, was indeed a deliberate choice. We agree that this aspect may be unclear to some readers, and we appreciate the suggestion. Reviewer 2 raised a similar point as well. We can revise Section 2.3 of the manuscript (VIC-CropSyst Model and Calibration) to clarify why we chose the VIC-CropSyst model rather than the standalone VIC model with text similar to what we have following two paragraphs.

This study is part of a larger, ongoing effort to understand the interplay between water supply, agricultural water availability, and the impacts of water shortages on agricultural productivity. The VIC-CropSyst model combines the VIC hydrology engine with the dynamic crop growth engine from the CropSyst model, enabling it to address both supply and demand sides of water usage in an integrated manner. This allows the model to capture how changes to water supply influence agricultural demand, and vice versa.

While the specific modifications related to the rain-snow partitioning scheme will be the same whether implemented in the VIC model or the VIC-CropSyst model, and the simulated impacts on snow and streamflow will be identical if the CropSyst crop growth engine is not invoked, implementing our changes in the VIC-CropSyst model offers significant advantages. It will enable us to address both standalone water supply applications and those related to the complex interaction between water supply and demand in future studies. Accurate snow simulations are crucial for the coupled VIC-CropSyst model, as snowmelt estimates directly affect soil moisture, water availability, and, consequently, crop growth and water demand.

We can update the manuscript to include this clarification and to better explain the rationale for using the VIC-CropSyst model.

    b)  From a water balance perspective, the analysis of snowmelt contribution to streamflow is interesting. However, given that there is not really meaningful observational constraint at SNOTEL sites, we are left only with model-to-model comparisons. While the authors are clear that this is the case, it is also possible that they could select a few watersheds within the region and compare simulated SWE/Q to the USDA's Basin Snow Water Equivalent estimates of SWE, normalized by runoff volume for the same watersheds. This represents some additional analysis, but it might provide a much more insightful comparison of the degree to which the change in precipitation partitioning influences whole-basin estimates of water supply.

We agree that comparing simulated SWE/Q to the USDA's basin estimates from observations, would provide additional insight, and can integrate this into a revised manuscript. This comparison would help to better assess the degree to which changes in precipitation partitioning influence observation-simulation matches of whole-basin water supply estimates. However, to ensure that observed flows are comparable to the VIC-CropSyst model outputs, which are "natural" flows without human influence, we focus on six watersheds (Boise River near Twin Springs (BOTWI), Clearwater River at Orofino (CLEAR), Coeur d'Alene above Shoshone creek near Prichard (COEUR), St. Joe River at Calder (JOECA), South Fork Payette River at Lowman (PAYLO), and Stehekin River at Stehekin (STEHE)) with minimal human influence as noted by USGS. For each basin, we follow the same methodology used by USDA in terms of the stream gauge and SNOTEL stations considered.

The analysis results for all simulation years across the six selected watersheds are provided below. Interestingly, we found that while dynamic partition decreases relative bias as compared to static partitioning and is almost always the best performing model for peak SWE (Figure R1 a) and annual Q ( Figure R1 b), that performance improvement does not translate to a better match

with observations for the SWE/Q ratio metric in all those cases. While dynamic partitioning is still the best method for the majority of cases for the ratio metric, it is the best model in fewer cases than for peak SWE and annual Q (the length of the vertical black lines in Figure R1).

[Figure]

*Figure R1. Relative model performance (RMP) chart for (a) peak SWE (SWE), (b) streamflow (Q), and (c) SWE/Q ratio, comparing the SRS (static rain-snow) and DRS (dynamic rain-snow) partitioning methods. The Y-axis is the fraction of stations for which a particular RMP is achieved, and X-axis is the difference between each model's mean absolute bias (MAB) and the best-performing model's MAB. The closer a model's curve is to the Y-axis and for longer, the better. The length of a model's curve exactly on the Y-axis indicates how frequently the model is best performing, and the distance of the curve from the Y-axis indicates how much worse a model's performance is relative to the best model. If the model under consideration is the best-performing one, the RMP value will be zero. If not, the RMP provides an indication of how far the model's performance is from the best-performing model.*

While this may seem counter intuitive, it is possible. We take the example of 2008 in the CLEAR watershed to understand why.

Observations:
      `peak SWE = 1047mm, annual Q = 9,640cfs, SWE/Q = 0.11`
Static Partitioning:
      `peak SWE = 845mm, annual Q = 7,890cfs, SWE/Q = 0.11`
      **`Absolute Relative Bias`**
      `peak SWE = 20%, annual Q = 18%, SWE/Q (mm/cfs) = 0%`
Dynamic Partitioning:
      `peak SWE = 891mm, annual Q = 9,437cfs, SWE/Q = 0.09`
      **`Absolute Relative Bias`**
      `peak SWE = 15%, annual Q = ~0%, SWE/Q (mm/cfs) = 18%`

In this case, with the dynamic partitioning, both peak SWE and annual Q got closer to the respective observations. However, the SWE/Q ratio got worse. This is because, in this case, annual Q improved a lot more with dynamic partitioning (18% absolute relative bias reduced to ~0%) in relative terms as compared to peak SWE (20% absolute relative bias reduced to 15%). Therefore, the ratio of the two worsened (0% absolute relative bias increased to 18%) as compared to the observed ratio. With the static partitioning, while there were larger errors in both peak SWE and annual Q than dynamic partitioning, the values were in the same ratio as observations with a better match of the ratio. This is a case where the original static method gave the right results for the wrong reason.

This is just one example, but there are other cases that lead to the static partitioning resulting in a better match with observations in spite of peak SWE and annual Q both showing a reduction in bias (blue color points in the lower darker quadrant of Figure R2) . But in most cases, the dynamic partitioning is still the best and when it is not, its performance is not that much lower than that of static partitioning (Figure R1 c).

[Figure]

*Figure R2: Change in bias for peak SWE, annual average Q and the ratio of the two. Each point corresponds to a year of simulation. Change in bias for the ratio of peal SWE to annual Q is indicated by the color of the point. Each panel corresponds to a watershed. The darker lower left quadrant of each panel, with negative values of change in bias on both the X- and Y-axis indicate years with reductions in both streamflow and SWE bias (better simulation-observation match) after the implementation of dynamic rain-snow partitioning. Red color of the points indicate reduction in bias (improvements in simulation-observation match) for the SWE/Q ratio, while the blue color represents years where the SWE/Q ratio simulation-observation match has degraded after applying the dynamic partitioning method.*

We will integrate this analysis and discussion into the manuscript.

**Minor Comments:**
Line 84: I would quibble slightly that the Columbia River Basin is entirely snow dominated, given that some significantly large areas within the basin are, in fact, rain dominated. I'd suggest slightly rewording this to "The CRB encompasses multiple states and portions of Canada in North America, and snow comprises a substantial fraction of annual precipitation in much of the watershed, particularly high mountain areas."

Yes, we agree. We can revise section 2.1 to add the suggested sentence.

Some of the figures (Figure 2, Figure 3) have some issues with the frames of the figure areas and seem misaligned with subfigures. I'd encourage the authors to address these minor formatting issues.

Yes, we can address this. Thanks for catching it.

For figure 9, the bin labels on the x-axis are slightly confusing. Is bin 4 really 10-20% bias? Or is it 0-20% bias? If the former, were there no results in the 0-10% bias range? Or is this what is meant by "Only bins with at least 8 stations are displayed." If so, in the text, please indicate that "Fewer than 8 SNOTEL stations fall within the range of a 0-10% change in bias, therefore we exclude this range in bias change from these box plots." For figure 9(d) the label of the plot could be more clear. I believe this is the fraction of precipitation to SWE, so perhaps P/SWE (-) would be clearer.

Yes, bin 4 corresponds to 10-20% bias and yes, it is excluded because there are fewer than 8 stations. We can modify the caption to clarify this. And yes, for figure 9(d), we agree that P/SWE (-) would be clearer, and we can make this change.

A modified Figure 9 from the original manuscript including these changes is pasted below.

[Figure]

*Figure 9. Percent bias in peak SWE and average temperature, elevation, and fraction of accumulative precipitation to peak SWE for bins of % change in bias in increments of 10% change in bias. Only bins with at least 8 stations are displayed and this corresponds to 84% of the stations. Bins 1, 2, and 3 with negative changes in bias correspond to performance improvements with DRS. Bin 4, with a positive change in bias corresponds to degradation in performance with DRS. Note that fewer than 8 SNOTEL stations fall within the range of a 0-10% change in bias, therefore we exclude this range in bias change from these box plots.*

*For each bin, the distribution of four aspects are provided across the stations within each bin. These include (a) biases of SRS and DRS methods, (b) average daily temperatures on wet days (precipitation > 1mm) during the snow season (November-April), (c) elevation, and (d) the ratio of cumulative precipitation from October 1st until peak SWE has been attained and the peak SWE value from SNOTEL observations. This is to get a sense of potential precipitation undercatch issues.*

---

## Author Comment (AC2)

**Reviewer #2**

Thank you for the feedback, suggestions for improvements, and for bringing the recent Wang et al. (2024) paper to our attention. Specific responses are provided in detail below and we have provided clarification related to the novelty of this work in comparison to other related work.

**Comment 1**

Singh et al. present a comprehensive analysis of changes to simulated snow cover evolution resulting from an updated dynamic rain-snow partitioning scheme in the VIC-CropSyst model in the Columbia River Basin (CRB) and other areas of the Pacific Northwest. They showed that a bivariate logistical regression method that predicts precipitation phase as a function of air temperature and relative humidity produces better simulated snow water equivalent (SWE) than the default VIC method that uses air temperature alone. The authors noted that improved snow outcomes led to a higher proportion of simulated streamflow coming from snowmelt versus the baseline case using the default VIC method.

I commend the authors on the hard work they put into this paper. The methods are clearly described, and results are straightforward and easy to follow. However, there are a few major shortcomings that I believe preclude this article from publication in HESS. I detail these below.

The first, and most important, issue is that the authors use the default VIC rain-snow partitioning scheme as the benchmark to which they compare the dynamic method. This default is a dual-threshold method with a lower, all-snow threshold of -0.5°C and an upper, all-rain threshold of 0.5°C, with mixed precipitation falling in between. Previous modeling work at multiple sites in the western US has shown this method to produce highly negative biases and low $r^2$ values in both SWE and snow depth (Jennings and Molotch, 2019). More recent observational work from the Sierra Nevada in the western US further highlighted the poor performance of the default VIC method compared to visual reports of rain, snow, and mixed precipitation. It was the second-worst partitioning method, only correctly predicting rain, snow, and mixed precipitation 47.1% of the time (Jennings et al., 2023). In other words, the VIC method got over half of its precipitation phase predictions wrong. Thus, the premise of the research—that a modified rain-snow partitioning scheme incorporating humidity and air temperature would improve on the default VIC method—is not a particularly evocative one. In fact, the researchers would be hard pressed to find a worse rain-snow partitioning method than the VIC default. This has the unsatisfying effect of producing results that are practically pre-ordained. In my opinion, this makes the work more appropriate as a case study for a different journal.

We appreciate the reviewer's feedback and acknowledge that there are many partitioning methods and would like to clarify our choice of benchmark. While we acknowledge that the default VIC rain-snow partitioning method has known limitations, we deliberately selected it because it reflects current practices in hydrological modeling. Despite its documented shortcomings, the dual-threshold method remains widely used in many hydrological models,

including VIC, and is a standard approach in the field (Harpold et al., 2017). Therefore, comparing the dynamic method to the default VIC method remains highly relevant, as it reflects the methods most commonly employed in hydrological modeling studies today.

Additionally, our intent was not to compare multiple partitioning methods, as in Jennings et al. (2023) or Jennings and Molotch (2019), but rather use this past comparison literature as a motivation, focus on a method that has been shown to work well in partitioning precipitation and evaluate whether and how using this method could improve the match between simulations and observations. In contrast to previous work, which has primarily focused on comparing the sensitivity of various snow and streamflow metrics to different partitioning methods, our goal was to select a partitioning method with a solid foundation in the literature and conduct a more thorough evaluation of the match between observations and simulations across a broader set of metrics than has been considered in prior studies. Following Reviewer 1's suggestion, we have expanded the analysis to include streamflow and the peak snow water equivalent (SWE)/annual average flow ratio metrics. Our aim was to understand where, when, and why we see improvements (or degradations) in the match between observations and simulations across a suite of metrics.

Furthermore, we cannot assume that improvements are preordained given the benchmark we chose. In fact, while we observe general improvements for snow magnitude metrics across most locations (with a few exceptions) by transitioning to a more dynamic precipitation phase partitioning, we also identified certain metrics (not addressed in the existing literature) where the match between observations and simulations actually degrades across most locations (see Table R1).

We can provide this clarity on the intent and contributions of this paper in the revised manuscript (drawing on Table R1 below) to make a case that our work offers more than just a case study applying known information to the Pacific Northwest.

**Comment 2**

The second major shortcoming is the novelty of the work. Wang et al. (2024) recently demonstrated similar findings when implementing the VIC model with a new rain-snow partitioning scheme (wet bulb temperature, TW) and comparing the outcomes to the default dual-threshold method. They showed "improved performance of the TW scheme in simulating snowfall fraction (SF) and snow water equivalent (SWE) in relation to in situ observations and a gridded SWE product." They also took the research a step further, analyzing the effect of method selection on simulated changes to snowpack and streamflow under future climate conditions.

Thank you for bringing the Wang et al. (2024) paper to our attention. Since it was published around the same time as our submission, we unfortunately missed it during our literature review process.

As noted earlier, the focus of our work was on comparing model simulations to observations, rather than analyzing the sensitivity of model outputs (in historical or future contexts) to different partitioning methods. We intentionally focused on a comprehensive comparison between observations and simulations, evaluating a broader set of metrics and a larger number of locations than is typical in the existing literature. In terms of evaluating simulation-observation matches, existing studies focused on a subset of precipitation phase, SWE magnitude, and snow depth, often reporting a general improvement (with some exceptions) in the match between simulations and observations when using partitioning methods that account for relative humidity.

In contrast, our study also includes several snow-timing-related metrics, where we found that the dynamic partitioning method led to a general degradation in the match between simulations and observations for some of the timing metrics in comparison with the default VIC approach. This highlights a situation where previous results were correct for the wrong reasons—improvements in precipitation-phase partitioning have revealed other model aspects that require further refinement. Additionally, given that Wang et al. (2024) primarily focused on the impact of the partitioning on future simulations, their discussion about observation-simulations comparisons are relatively limited while our work takes a more comprehensive look into that aspect. Also, while Wang et al. (2024) note observation-simulation mismatches for annual mean SWE in high elevation regions in the Colorado River basin, our results in the Columbia River basin are different with mismatches spanning all elevation ranges. It would be interesting to compare the results across the regions and relate them to more explainable physiographic differences.

We believe that our study offers insights beyond a simple case study of the Pacific Northwest and contributes new findings that go beyond reiterating previous work. To clarify how our research fits within the broader literature, we have provided a table that highlights key comparisons with existing studies and underscores the novel aspects of our work.

We agree that Wang et al. (2024) and Jennings et al. (2023) should be cited in our study. Drawing on Table R1 below, we will revise the introduction and discussion sections to better clarify the specific contributions of our work and how it builds on prior research.

Table R1: Comparison of our paper and literature reference here

| Paper | Geographic context | Historical timeframe | Metrics for observation-simulation match | Outcome of incorporating a RH-based dynamic partitioning on simulation-observation match |
|---|---|---|---|---|
| Jennings and Molotch, 2019 | 11 experimental stations in Western US. | 2004-2011 | daily SWE

snow depth | Mostly improvements. |
| Jennings et al., 2023 | Lake Tahoe region in California. | 2020-2021 | precipitation phase | Mostly improvements. |
| Wang et al., 2024 | Colorado River basin (124 SNOTEL stations and a gridded observational product) | 1971-2018 (varies by station; at least 20 years needed) | average annual snowfall fraction

annual average SWE magnitude

streamflow at the basin outlet | Mostly improvements except for annual average SWE in high elevations(>3500m). |
| This paper | Columbia River basin (164 SNOTEL stations) | 1996-2015 | peak SWE magnitude

peak SWE timing

daily SWE

snow start date

snow off date

snow duration

streamflow *

$SWE_{peak}/Q_{annual}$ * | Mostly improvements for snow magnitude metrics with some exceptions. This is similar to Wang et al. (2024) except that our contexts for degradation in SWE magnitude match are unrelated to elevation unlike Wang et al.(2004). See Figure 9 in our paper.

Mostly degradation in match for *snow off* and *snow duration* metrics (with some exceptions) which is not noted in the literature to the best of our knowledge.

There is a mixed direction of response for the newly added SWE/Q ratio metric as well. |

\* The last two metrics are newly added in response to reviewer #1's comments so that we can have basin-scale simulation-observation comparison metrics.

**Comment 3**

A minor issue I had with the work was the use of VIC-CropSyst versus VIC. It was never made clear the motivation for using the coupled crop model when focusing on the mountain regions of the CRB. The authors did note "While the study primarily focuses on snow dynamics simulated by the VIC component of the model, the full VIC-CropSyst model was used because

the streamflow calibration was performed on the coupled model version." This sounds like maybe the baseline simulations using VIC-CropSyst were already on hand, so the authors used the same coupled setup for the dynamic method for easier comparisons. Additionally, the authors said the model was calibrated for "five soil parameters" despite their focus on snow outcomes. This is perplexing to me.

Thanks for raising this lack of clarity. The sentence we had was a bit misleading in terms of why we used the VIC-CropSyst model. The use of VIC-CropSyst in this study, rather than the standalone VIC model, was a deliberate choice. Reviewer 1 raised a similar point and provided suggestions to revise Section 2.3 of the manuscript (VIC-CropSyst Model and Calibration) to help clarify why we chose the VIC-CropSyst model rather than the standalone VIC model. See related text in the following two paragraphs.

This study is part of a larger, ongoing effort to understand the interplay between water supply, agricultural water availability, and the impacts of water shortages on agricultural productivity. The VIC-CropSyst model combines the VIC hydrology engine with the dynamic crop growth engine from the CropSyst model, enabling it to address both supply and demand sides of water usage in an integrated manner. This allows the model to capture how changes to water supply influence agricultural demand, and vice versa.

While the specific modifications related to the rain-snow partitioning scheme will be the same whether implemented in the VIC model or the VIC-CropSyst model, and the simulated impacts on snow and streamflow will be identical if the CropSyst crop growth engine is not invoked, implementing our changes in the VIC-CropSyst model offers significant advantages. It will enable us to address both standalone water supply applications and those related to the complex interaction between water supply and demand in future studies. Accurate snow simulations are crucial for the coupled VIC-CropSyst model, as snowmelt estimates directly affect soil moisture, water availability, and, consequently, crop growth and water demand.

We can update the manuscript to include this clarification.

Regarding calibration, it is typical for hydrology model calibration to focus on just streamflow. Multi-objective calibration is of interest to us as a next step and we have commenced on it, but it is an involved effort as noted in the response to comment 4. 3 below and outside the scope of this work.  Similar to Wang et al. (2024) which also did not calibrate for snow metrics, we can clarify in the manuscript text that performance improvements noted in the study can be further enhanced via calibration of snow metrics. Performance degradations could potentially be alleviated as well.

**Comment 4**

I would again like to acknowledge the effort the authors put into this manuscript. It is not easy to wrangle this amount of data and write a straightforward, clearly described manuscript. While I don't believe the paper in its current form is suitable for publication in HESS, I do think its appropriateness could be enhanced with a few modifications:

Thanks for the feedback and suggestions. We have responded to each point below.

**Comment 4.1**

There is a lot of work out there that makes large conclusions about hydroclimatology in the western US using VIC and its now demonstrably poor default rain-snow partitioning method. This work has been published in top-tier journals like *Nature* and *Water Resources Research*, to name just two. The authors may wish to reframe their paper around this issue. Have our previous VIC simulations misled us about the hydroclimatology of the Pacific Northwest? What does this mean for water resources modeling and hydrologic forecasting? Do we understand the potential impacts of climate change on snow cover evolution and streamflow?

We agree that these are important questions. This analysis is something we are currently working on and a manuscript is under preparation. However, as noted in the earlier comment, we believe that the comprehensive observation-simulation comparisons performed in this work provide new contributions worthy of publication.

**Comment 4.2**

Reconsider the use of VIC-CropSyst. If the purpose is to compare snow simulations in mountain regions, it is hard to justify the coupled model versus VIC alone.

We agree that the motivation for using the VIC-CropSyst model was unclear. Please see our related response to comment 3 above.

**Comment 4.3**

Similarly, calibrate the model's snow parameters and see what happens.

Multi-objective calibration (for streamflow, snow, and evapotranspiration) is not common, is an important step and one we are exploring. This is an involved undertaking, given that the computational complexity is much larger. This is also why we are considering the VIC-CropSyst model so that actual evapotranspiration from croplands can also be considered in snowmelt watersheds that have a predominant agricultural land-use. This is an ongoing effort and we will work on a related manuscript when the effort is complete.

**Comment 4.4**

Consider other rain-snow partitioning methods. Beating the default VIC method, as demonstrated above, is no challenge given its poor track record. What about spatially variable air temperature thresholds, wet bulb temperature thresholds, etc.?

As noted in earlier responses, the motivation for our study was a comparison between observations and simulations. Rather than comparing multiple partitioning approaches, we focused on adopting a method suggested in the literature as performing well and concentrated on observation-simulation comparisons. While we are open to comparing different methods, we do not expect such comparisons to add significant value to the research question we are addressing as described below.

Regarding spatially-varying, temporally-constant temperature thresholds for rain-snow partitioning, we believe these approaches will likely be insufficient because they do not account for the temporal variability in relative humidity (Figure R1) and therefore surface air temperature thresholds for partitioning (Figure R2).

Figure R2 isolates the temporal variation in the surface air temperature threshold for partitioning precipitation into 50% rain and 50% snow based on our reported results for each grid. Across all bins, there is clear temporal variation (though the range of temporal variation can vary by grid as shown in the box plot range), suggesting that spatially-varying but temporally-constant surface air temperature thresholds would not be an effective strategy.

[Figure]

*Figure R1: Frequency of grids with different temporal differences in RH (difference between 95th and 5th percentile RH across wet days (precipitation > 1mm) between November through April and years). We can see that most grids have a temporal difference between 27% to 33% which is large enough difference to impact temporal differences in snow fraction estimates.*

[Figure]

*Figure R2. Temporal variation in the surface air temperature threshold for partitioning precipitation into 50% rain and 50% snow. For each grid and wet day (precipitation > 1 mm) during the snow period (November-April), the probability of snowfall is computed using binary logistic regression method*

*(Jennings et al., 2018). The 50-50 rain-snow temperature threshold is calculated annually for each grid by RH bins. Temporal variation in this threshold is then determined by calculating the difference between the 5th and 95th percentile threshold values across the years. Each boxplot shows the distribution of threshold differences across multiple grids. For example, in the (40,50] RH bin, the median grid shows a temporal variation in the 50-50 rain-snow threshold of approximately 1.6°C, based on the 5th and 95th percentile values.*

RH-based approaches like those implemented by Jennings et al. (2018), or using the wet-bulb temperature (Tw) as in Wang et al. (2024), or utilizing static Tw thresholds, would account for both spatial and temporal heterogeneity providing solutions as noted in the literature. Our initial hypothesis was that observation-simulation matches across multiple snow and streamflow metrics would be similar across all these RH-based approaches, in line with the generally comparable performance of these methods in simulating precipitation phase, as shown in studies like Jennings and Molotch (2019) or others that compare numerous partitioning methods. Thus, our hypothesis was that our basic findings would likely remain consistent across any method that accounts for air temperature and relative humidity in some form.

In response to this comment, we implemented a comparison between the wet-bulb temperature approach (Wang et al., 2024) and the approach used by Jennings et al. (2018), and as expected, we did not observe a major difference in performance for the metrics we considered (Figures R3, R4 and R5). The yellow and orange lines in these figures corresponding to the two RH-based methods are almost overlaid and similar.

We can add these figures as an appendix to the manuscript and note the consistency in performance in the results/discussion section of the manuscript.

[Figure]

*Figure R3. Relative Model Performance (RMP) chart for peak SWE (a) magnitude, and (b) timing, comparing SRS (static rain-snow), DRS (dynamic rain-snow), and DRS_TW (dynamic rain-snow based on wet bulb temperature) partitioning methods. The Y-axis is the fraction of stations for which a*

*particular RMP is achieved, and the X-axis is the difference between each model's mean absolute bias (MAB) and the best-performing model's MAB. The closer a model's curve is to the Y-axis and for longer, the better. The length of a model's curve exactly on the Y-axis indicates how frequently the model is best performing, and the distance of the curve from the Y-axis indicates how much worse a model's performance is relative to the best model. If the model under consideration is the best-performing one, the RMP value will be zero. If not, the RMP provides an indication of how far the model's performance is from the best-performing model.*

[Figure]

*Figure R4. Relative Model Performance chart comparing SRS , DRS , and DRS_TW partitioning methods for snow phenology metrics i.e., (a) snow-start, (b) snow-off, and (c) snow-duration . See Figure R3 caption for more details on interpretation.*

[Figure]

*Figure R5. Relative Performance chart for NSE of daily SWE (see Figure R3 caption for more details on interpretation), comparing SRS , DRS , and DRS_TW partitioning methods. Model performance (NSE) of daily SWE for the snow season (November through April)*

**Comment 4.5**

I thank the authors for their time and wish them luck with their manuscript!

Thank you for your feedback and for bringing Wang et al. (2024) and other suggestions to our attention. We appreciate the feedback.

**References**

Harpold, A. A., Kaplan, M. L., Klos, P. Z., Link, T., McNamara, J. P., Rajagopal, S., Schumer, R., and Steele, C. M.: Rain or snow: hydrologic processes, observations, prediction, and research needs, Hydrol. Earth Syst. Sci., 21, 1–22, https://doi.org/10.5194/hess-21-1-2017, 2017.

Jennings, K.S. and Molotch, N.P. (2019). The sensitivity of modeled snow accumulation and melt to precipitation phase methods across a climatic gradient. Hydrology and Earth System Sciences. https://doi.org/10.5194/hess-23-3765-2019

Jennings, K.S., Arienzo, M.A., Collins, M., Hatchett, B., Nolin, A.W., and Aggett, G.R. (2023). Crowdsourced Data Highlight Precipitation Phase Partitioning Variability in Rain-Snow Transition Zone. Earth and Space Science. https://doi.org/10.1029/2022EA002714

Wang, Z., Vivoni, E. R., Whitney, K. M., Xiao, M., & Mascaro, G. (2024). On the sensitivity of future hydrology in the Colorado River to the selection of the precipitation partitioning method. *Water Resources Research*, *60*(6), e2023WR035801. https://doi.org/10.1029/2023WR035801

Jennings, K. S., Winchell, T. S., Livneh, B., and Molotch, N. P.: Spatial variation of the rain–snow temperature threshold across the Northern Hemisphere, Nat. Commun., 9, 1148, 2018.